# Prescription of anti-influenza drugs in Japan, 2014–2020: A retrospective study using open data from the national claims database

**Akahito Sako** [1]*, **Yoshiaki Gu**[2], **Yoshinori Masui**[1], **Kensuke Yoshimura**[3], **Hidekatsu Yanai**[1], **Norio Ohmagari**[4]

**1** Department of Internal Medicine, Kohnodai Hospital, National Center for Global Health and Medicine, Ichikawa, Chiba, Japan, **2** Department of Infectious Diseases, Tokyo Medical and Dental University Graduate School of Medical and Dental Sciences, Bunkyo-ku, Tokyo, Japan, **3** Center for Next Generation of Community Health, Chiba University Hospital, Chuo-ku, Chiba, Japan, **4** Disease Control and Prevention Center, Center Hospital, National Center for Global Health and Medicine, Shinjuku-ku, Tokyo, Japan

* dsako@hospk.ncgm.go.jp

## Abstract

### Background

Both physicians and patients are proactive towards managing seasonal influenza in Japan and six drugs are approved. Although many countries have national influenza surveillance systems, data on nationwide prescription practices of anti-influenza drugs are lacking. Therefore, we aimed to clarify the status of anti-influenza drug use in Japan by analyzing real-world data.

### Methods

This retrospective study analyzed open data from the National Database of Health Insurance Claims and Specific Health Checkups, which covers most claims data from national health insurance. We estimated the annual number of patients prescribed anti-influenza drugs, which drugs they were prescribed, the patients' age and sex distribution, drug costs, and regional disparities for the period 2014–2020.

### Results

For 2014–2019, an estimated 6.7–13.4 million patients per year were prescribed anti-influenza drugs, with an annual cost of 22.3–48.0 billion JPY (Japanese Yen). In addition, 21.1–32.0 million rapid antigen tests were performed at a cost of 30.1–47.1 billion JPY. In 2017, laninamivir was the most frequently prescribed anti-influenza drug (48%), followed by oseltamivir (36%), while in 2018, the newly introduced baloxavir accounted for 40.8% of prescriptions. After the emergence of COVID-19, the estimated number of patients prescribed anti-influenza drugs in 2020 dropped to just 14,000. In 2018, 37.6% of prescriptions were for patients aged < 20 years compared with 12.2% for those aged $\geq$ 65 years. Prescriptions for inpatients accounted for 1.1%, and the proportion of prescriptions for inpatients increased

**Data Availability Statement:** The datasets used and/or analyzed in this study are publicly available

(https://www.mhlw.go.jp/stf/seisakunitsuite/bunya/0000177182.html).

**Funding:** AS received a grant from the National Center for Global Health and Medicine (Grant Numbers: 19A1026, 20A3001, and 23A3002). The funders had no role in study design, data collection and analysis, decision to publish, or preparation of the manuscript.

**Competing interests:** The authors have declared that no competing interests exist.

**Abbreviations:** NDB, National Database of Health Insurance Claims and Specific Health Checkups; COVID-19, Coronavirus disease 2019; NESID, National Epidemiological Surveillance for Infectious Diseases; JPY, Japanese Yen; FY, fiscal year.

with age, with men were more likely than women to be prescribed anti-influenza drugs while hospitalized.

## Conclusions

Based on our clarification of how influenza is clinically managed in Japan, future work should evaluate the clinical and economic aspects of proactively prescribing anti-influenza drugs.

## Background

Seasonal influenza is a common viral infection prevailing in winter. Although the incidence and significance of seasonal influenza has decreased since the emergence of coronavirus disease 2019 (COVID-19) [1,2], its clinical, social, and economic impacts remain profound [3–5]. Currently, six kinds of drugs are approved and available for seasonal influenza in Japan—amantadine, oseltamivir, zanamivir, laninamivir, peramivir, and baloxavir—even though with the exception of oseltamivir and zanamivir, there is insufficient evidence of their effectiveness. Favipiravir is also approved but limited for novel or re-emerging influenza virus infections. Anti-influenza drugs are commonly prescribed in Japan, and both physicians and patients are proactive toward using anti-influenza drugs, even non-elderly patients without underlying disease [5–7]. Indeed, it has been reported that Japan consumes more anti-influenza drugs than any other country [8–11]. However, nationwide or large-scale data on the prescription practices of anti-influenza drugs are lacking in both Japan and other countries.

The National Institute of Infectious Diseases reports nationwide data on influenza surveillance, including annual estimates of influenza cases, deaths by influenza, seasonal trends, and distribution by region and age, but not prescription data [2]. Surveillance data are based on reports from about 5,000 sentinel hospitals and clinics, accounting for about one-tenth of the pediatric and internal medicine facilities nationwide, and concerns have been raised about sampling bias and underreporting by physicians [12].

In this study, we investigated the clinical epidemiology of the prescription practices of anti-influenza drugs for seasonal influenza in Japan by analyzing a nearly complete enumeration dataset from the national health insurance claims database.

## Methods

### National database open data

This is a retrospective cohort study using open data from Japan's National Database of Health Insurance Claims and Specific Medical Checkups (NDB). Japan has a population of 126 million and a universal health care system. The Ministry of Health, Labour and Welfare has used the NDB to collect almost all national health insurance claims for both inpatients and outpatients since 2009. The NDB contains information such as patient sex, age, diagnosis, procedures and surgeries, and the dates and doses of prescriptions issued. NDB Open Data available from the NDB Open Data website does not provide individual data but rather aggregate data, including annual numbers of surgeries, medical services, tests, and prescriptions in Japan, stratified by sex, 5-year age intervals, inpatient/outpatient status, and geographic location of clinics or hospitals in Japan's 47 prefectures [13]. In Japan, the fiscal year (FY) runs from April 1 to March 31, and as of September 2022, data are available for FY2014 to FY2020. We analyzed data for this entire period with a focus on FY2018 in order to describe clinical practice

before the emergence of COVID-19 and the impact of the then newly approved baloxavir. As a denominator, we used the population of Japan stratified by age, sex, and prefecture, as reported by the Statistics Bureau of Japan [14].

Several studies have examined data from the NDB [15–17] and NDB Open Data [18,19]. The need for informed consent was waived because the data published by NDB Open Data are anonymized and publicly available. The institutional review board of the National Center for Global Health and Medicine reviewed and approved the study protocol.

## Anti-influenza drugs and tests for influenza

We investigated the annual prescription of five anti-influenza drugs for seasonal influenza: oseltamivir, zanamivir, laninamivir, peramivir, and baloxavir. We did not include amantadine because we consider almost all prescriptions of amantadine to be for Parkinson's disease rather than influenza. Data are available on the annual prescriptions of the top 100 anti-viral drugs, including the generic equivalents, dosage forms, whether the drug was administered orally, via injection, or via inhalation, and whether the patient was an inpatient or outpatient; for 2014, these data were available for only the top 30 anti-viral drugs. Zanamivir has been available in Japan since 2000, oseltamivir since 2001, laninamivir and peramivir since 2010, and baloxavir since 2018. Between 2007 and 2018, oseltamivir was contraindicated for patients aged 10–19 years in Japan because some cases of neuropsychiatric adverse effects and falls were reported [7]. Data on prophylactic use of anti-influenza drugs are not included in NDB because such usage is not covered by national health insurance.

In Japan, a prescription must comply strictly with the dosage and duration stated in the package insert, otherwise, the cost of the drug will not be reimbursed by national health insurance. Unlike the antibiotics prescribed for bacterial infection, the daily dose and duration of anti-influenza drugs are basically fixed; for example, oseltamivir is administered as a 75-mg capsule twice daily for 5 days for patients aged > 9 years. Based on the package insert, we considered the following prescriptions to indicate 1 influenza patient: oseltamivir, 750 mg (capsules or dry syrup) as 1 influenza patient aged > 9 years and oseltamivir 20 mg/kg as 1 influenza patient aged ≤ 9 years; zanamivir, 20 inhalations; laninamivir, one inhalation as 1 influenza patient aged 0–9 years and two inhalations as 1 influenza patient aged > 9 years or 1 bottle of inhalation suspension; peramivir, one infusion of 300 mg or 150 mg; and baloxavir, 10 mg as 1 patient aged 0–4 years, 20 mg as 1 patient aged 5–9 years, 30 mg as 1 patient aged 10–14 years, and 40 mg as 1 patient aged >14 years. For pediatric patients, we used the mean body weight reported for each age by the 2017 School Health Statistics [20] and the 2010 National Growth Survey on Preschool Children [21]. We assumed a mean body weight of 12.0 kg and 24.4 kg for boys aged 0–4 years and 5–9 years, respectively, and corresponding values of 11.5 kg and 23.9 kg for girls.

Because NDB Open Data protects personal privacy through anonymization, there are some missing values in the data. For example, data are made available in the form of an Excel spreadsheet on the total annual prescriptions of oseltamivir 75 mg capsules and distribution by sex, 5-year age interval, and prefecture, but if the number of prescriptions in a specific cell is low, the data are masked. The cut-off values for masking are 1,000 tablets, capsules, blister packs, kits, or bottles and 1,000 g for dry syrup. For infusion bags or vials, the cut-off value was 1,000 in 2015 and has been 400 since 2016. For example, if only 450 oseltamivir capsules were prescribed for boys aged 0–4 years in 2019, this information would be masked. Moreover, even if only one cell contained a value below the cut-off, until 2017 the entire row was masked, and thereafter the cells with the two lowest values are masked. When we calculated the total

number of patients prescribed anti-influenza drugs, we assumed that the prescriptions with unknown age group were adult doses.

Age distribution by sex is available, but the age distribution within each prefecture is not. When we calculated the total number of patients prescribed anti-influenza drugs in each prefecture, we extrapolated the age distribution based on that of prescriptions nationwide.

Rapid antigen tests, serologic tests, and nucleic acid amplification tests are available in Japan but are performed only when ordered by a physician. Self-tests and over-the-counter retail tests were not approved in Japan until 2022. However, we calculated only rapid antigen tests, because only 14,759 serologic tests and 84 nucleic acid amplification tests were performed in 2019.

In terms of the costs associated with anti-influenza drugs and rapid antigen tests, we used only the reimbursement fees for drugs and tests and did not include the fees for physicians and pharmacists or for hospitalization. For our calculations, we assume 1 USD is equivalent 110 JPY during the study period. Accordingly, costs were calculated as follows: rapid antigen test, 1,430 JPY; one treatment course of zanamivir for adults, 34,700 JPY in 2014 and 29,420 JPY in 2019; one treatment course of oseltamivir sold under a brand name, 31,790 JPY in 2014 and 27,200 JPY in 2019, and as a generic drug, 13,600 JPY in 2019; one treatment course of laninamivir, 42,800 JPY; single infusion of peramivir 300 mg 62,160 JPY; and one treatment course of baloxavir, 47,890 JPY. The reimbursement price for a drug is set by the government and is gradually reduced every two years. The consumer price index was stable in Japan during study period. It was highest in 2019 (100.0) and lowest in 2014 (97.4); therefore, we did not adjust the costs of drugs for inflation.

## Results

### Annual number of patients and cost of anti-influenza drugs and rapid antigen tests

The estimated annual number of patients prescribed anti-influenza drugs in 2014–2019 ranged from 6.7 to 13.4 million (Fig 1). The annual number of rapid antigen tests performed in the same period was 21.3, 21.5, 24.6, 32.0, 24.8, and 21.1 million times, respectively. In 2017, the most prescribed anti-influenza drug was laninamivir (48.1%), followed by oseltamivir (36.2%), zanamivir (11.4%), and peramivir (4.4%). In 2018, the newly introduced baloxavir accounted for 40.8% of prescriptions, followed by oseltamivir (28.7%), laninamivir (22.9%), zanamivir (4.7%), and peramivir (2.8%). In 2019, baloxavir dropped to account for just 15.9%. After the emergence of COVID-19, the estimated number of patients prescribed anti-influenza drugs was only 14,000 in 2020 and 1.7 million rapid antigen tests were performed.

The annual cost of anti-influenza drugs in 2014–2019 ranged from 22.3 to 48.0 billion JPY and that of rapid antigen tests in the same period was 31.7, 32.0, 36.1, 47.1, 35.5, and 30.1 billion JPY, respectively (Fig 2). Until 2017, the most expensive drug was laninamivir (54.3% of the total expenditure on anti-influenza drugs in 2017), followed by oseltamivir (28.9%), zanamivir (9.7%), and peramivir (7.2%). In 2018, baloxavir accounted for 49.5%, followed by laninamivir (25.6%), oseltamivir (16.6%), peramivir (4.5%), and zanamivir (3.9%). The only generic form of an anti-influenza drug available in Japan is oseltamivir, which became available in 2018. The proportion of prescriptions for the generic forms of oseltamivir capsule and dry syrup was 36.6% and 30.0% in 2018 and 53.0% and 42.1% in 2019, respectively.

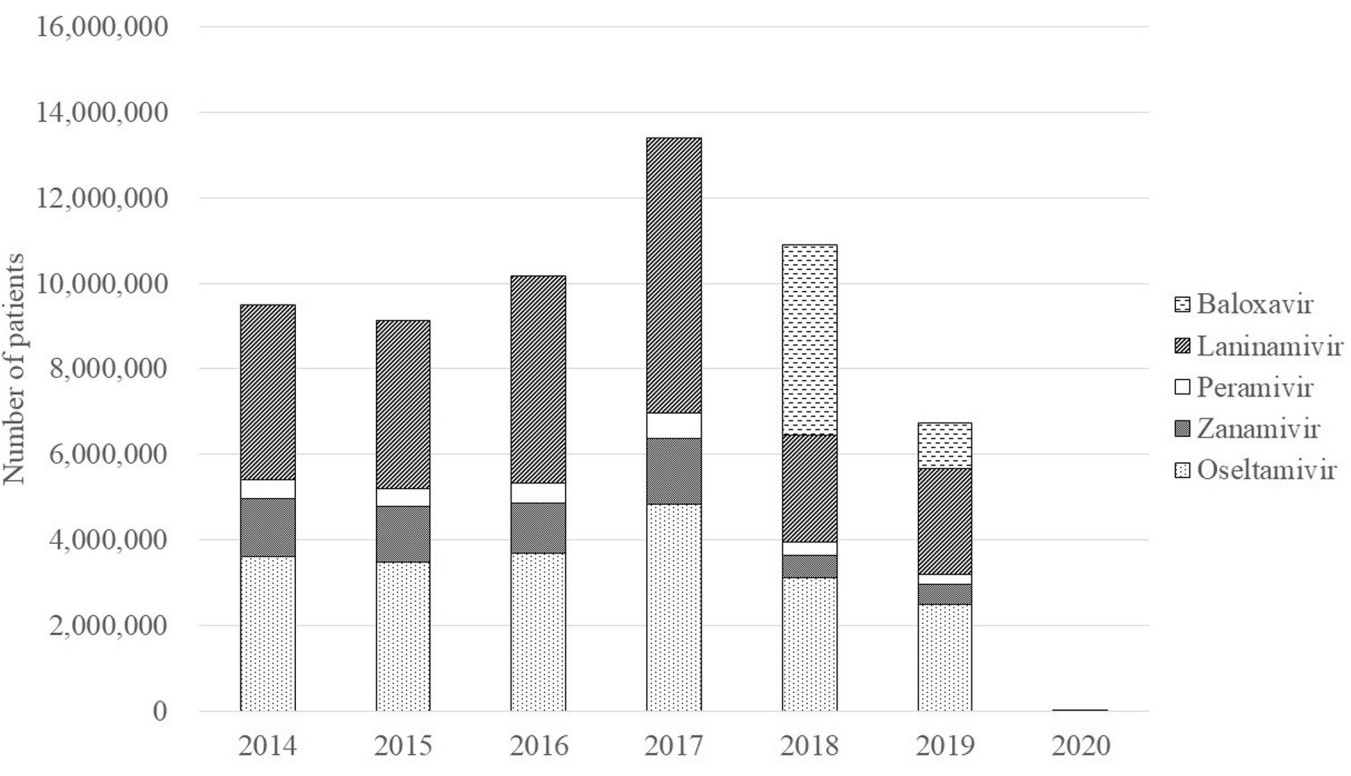

**Fig 1. Annual number of patients prescribed anti-influenza drugs in Japan (2014–2020).**

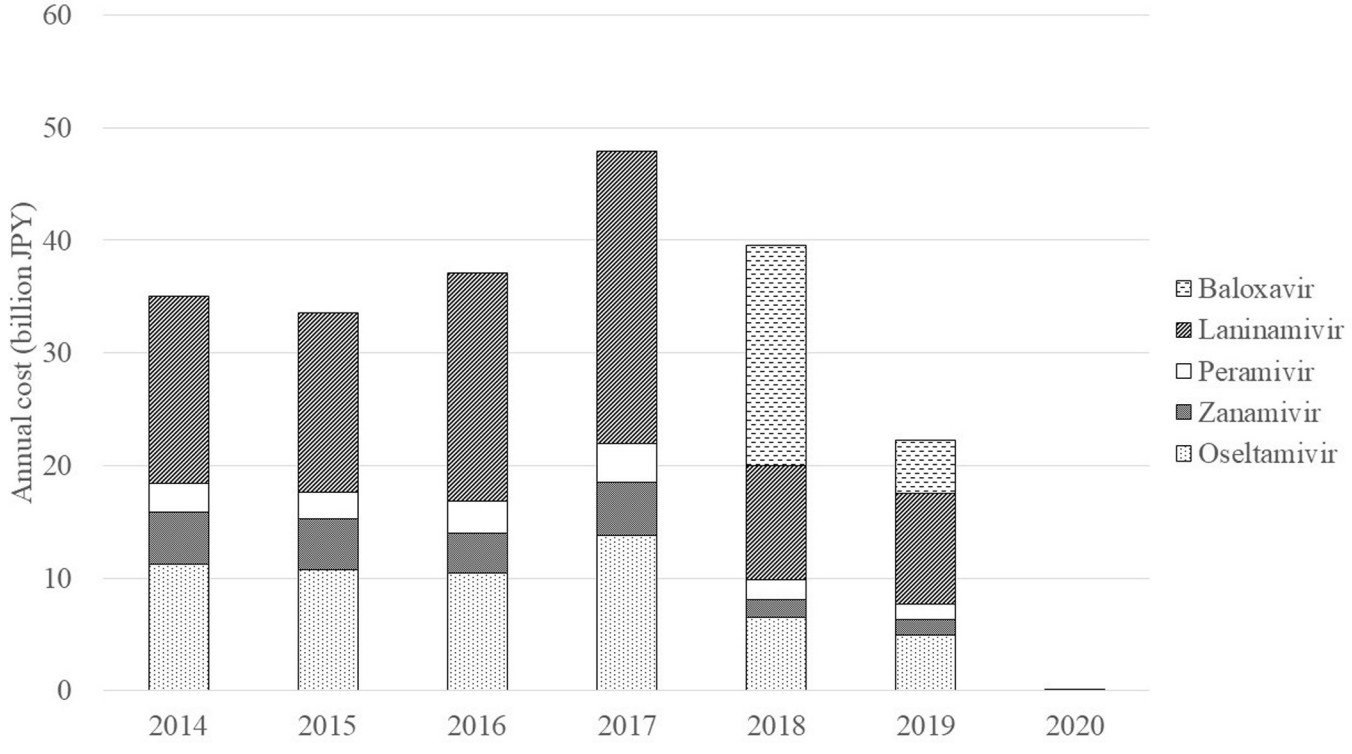

**Fig 2. Annual cost of anti-influenza drugs in Japan (2014–2020).**

## Age and sex distribution

In 2018, the sex distribution of patients prescribed anti-influenza drugs was nearly even, with female patients accounting for 50.5% of the total (Fig 3). Males were predominant among patients aged < 20 years, and females were predominant among patients aged ≥ 25 years. Among both sexes, most patients were aged 5–9, followed by those aged 10–14 and 0–4, with a second peak seen in those aged 35–49. Patients aged < 20 accounted for 37.6% of the total, while patients aged ≥ 65 accounted for 12.2%.

With the general population as the denominator, we found the highest annual incidence of anti-influenza drug prescriptions in aged 5–9, at 259 per 1,000 population for males and 245 for females (Fig 4). The incidence was higher for males aged 0–19 and for females aged 20–69.

In 2018, the most frequently prescribed drug for patients aged < 10 was oseltamivir (Fig 5) and that for patients aged 10–94 was baloxavir. Laninamivir was the second most frequently prescribed drug for patients aged 10–59. Zanamivir was relatively frequently prescribed for patients aged 5–19, and peramivir was frequently prescribed for patients aged ≥ 75.

## Geographical differences by prefecture

In 2018, the number of patients prescribed anti-influenza drugs per 1,000 population was highest in Kumamoto (97.9), followed by Gifu (96.8) and Aichi (96.4) and was lowest in Okinawa (65.3), Shimane (66.4), and Aomori (66.9) (Fig 6).

Choice of prescription varied among the 47 prefectures (Fig 7). Baloxavir was the most frequently prescribed drug in 39 prefectures, the second most in 6 prefectures, and the third most in 2 prefectures. Baloxavir accounted for the highest proportion in Aichi (50.7%) and the lowest in Okinawa (26.1%). Oseltamivir was the most frequently prescribed drug in 6 prefectures, the second most in 38 prefectures, and the third most in 3 prefectures. Oseltamivir accounted for the highest proportion in Okinawa (40.4%) and the lowest in Gifu (22.1%). Laninamivir was the most frequently prescribed drug in 2 prefectures, the second most in 3 prefectures, and the third most in 42 prefectures. Laninamivir accounted for the highest proportion in Shimane (35.1%) and the lowest in Mie (15.2%). Zanamivir was the least frequently prescribed drug in 9 prefectures and accounted for the highest proportion in Yamanashi (8.7%) and the lowest in Shiga (2.1%). Peramivir was the least frequently prescribed drug in 38 prefectures and accounted for the highest proportion in Iwate (6.5%) and the lowest in Saga (0.7%).

## Inpatient and outpatient prescriptions

Prescriptions for inpatients accounted for 1.1% of the total (Fig 8). The proportion of prescriptions for inpatients increased with age. Male were more likely to be prescribed anti-influenza drugs for inpatient than female, regardless of age. The most frequently prescribed drug for inpatients was peramivir (61.8%), followed by oseltamivir (28.2%), baloxavir (7.1%), and laninamivir (3.0%). Peramivir was the most frequently prescribed drug among male inpatients aged < 15 and ≥ 65 years; data for 15–29 year olds were not available due to the small number of inpatients (Fig 9). Peramivir was the most frequently prescribed drug among female inpatients aged < 10 and ≥ 65 years; data for 10–24 year olds were not available due to the small number of inpatients (Fig 10). Peramivir for inpatients accounted for 25.1% of all peramivir prescriptions.

The proportion of inpatient prescriptions varied among prefectures. Hokkaido had the highest proportion (2.1%), followed by Iwate (1.6%) and Toyama (1.5%). Yamanashi and Wakayama had the lowest proportion (0.2% each). In Shimane and Tottori, inpatient prescription data were not available due to the small number of inpatients. Peramivir was the most

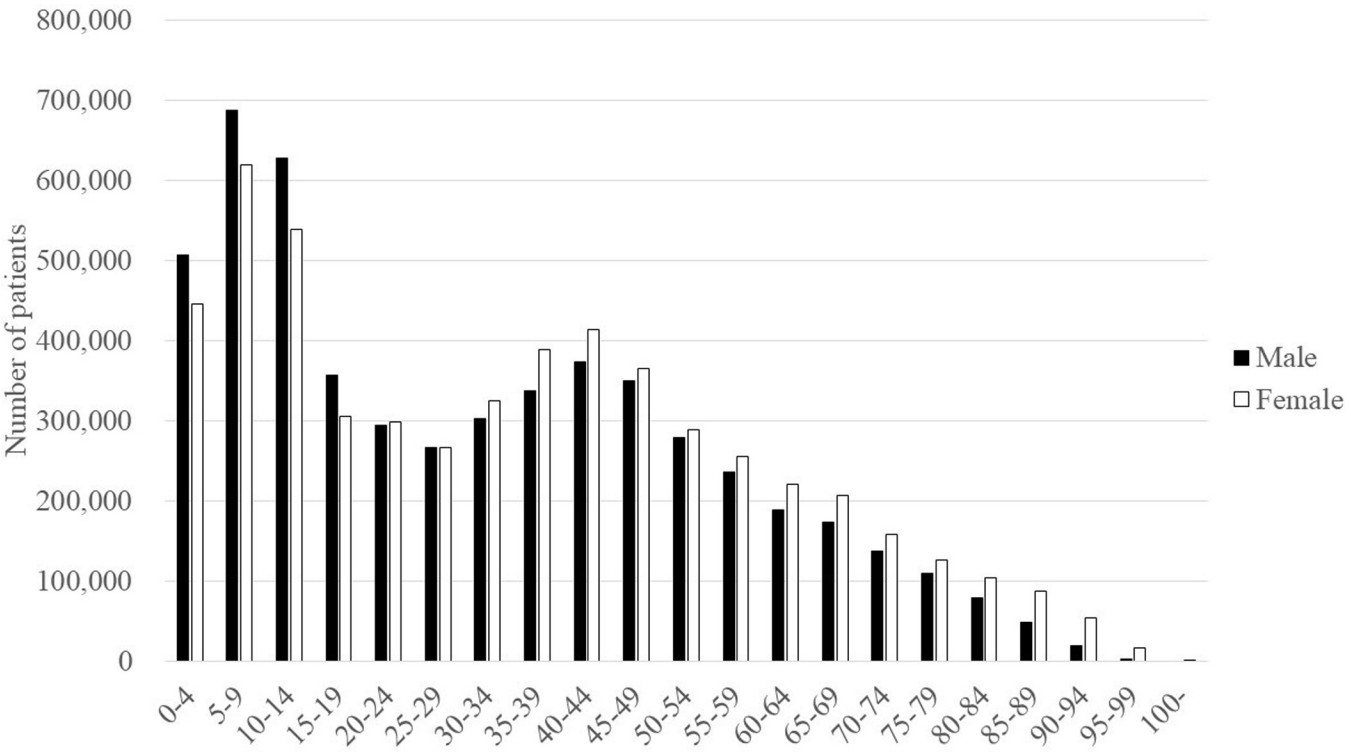

**Fig 3. Age and sex distribution of patients prescribed anti-influenza drugs in 2018.**

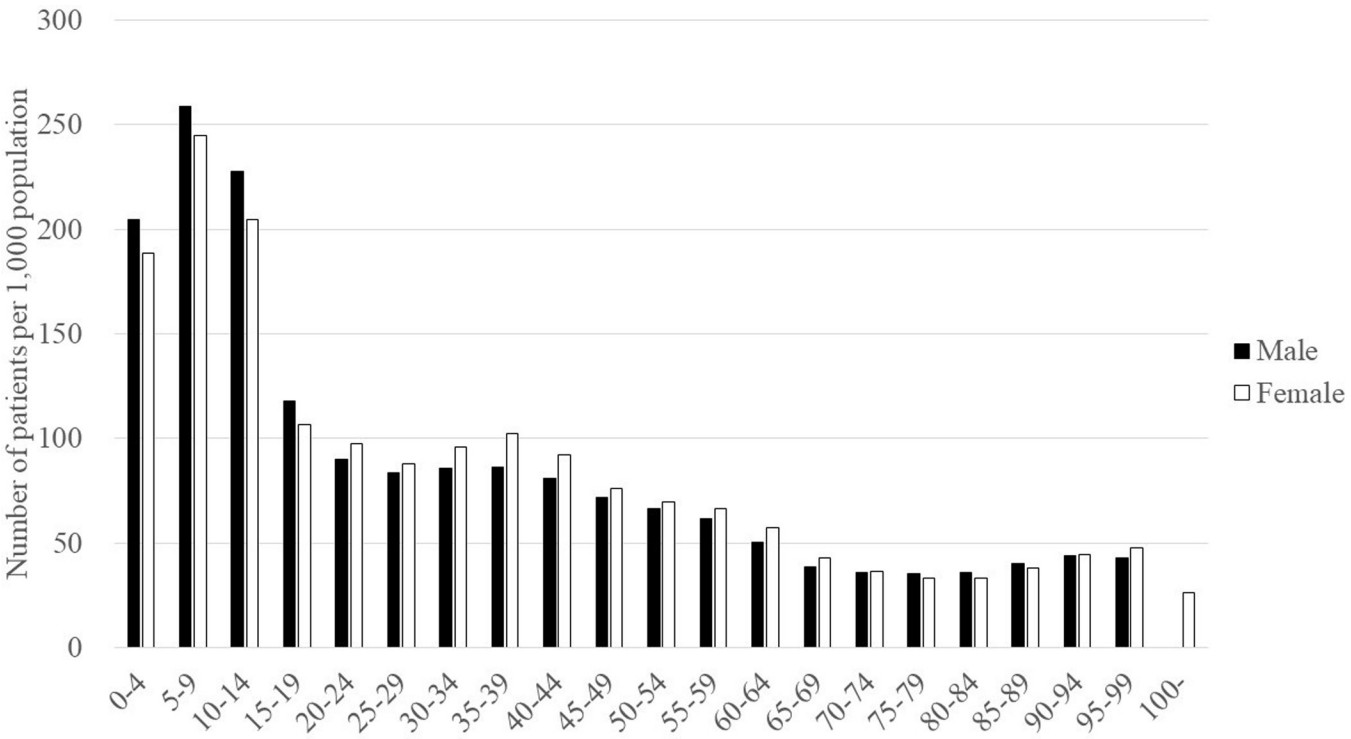

**Fig 4. Age and sex distribution of patients prescribed anti-influenza drugs per 1,000 population in 2018.**

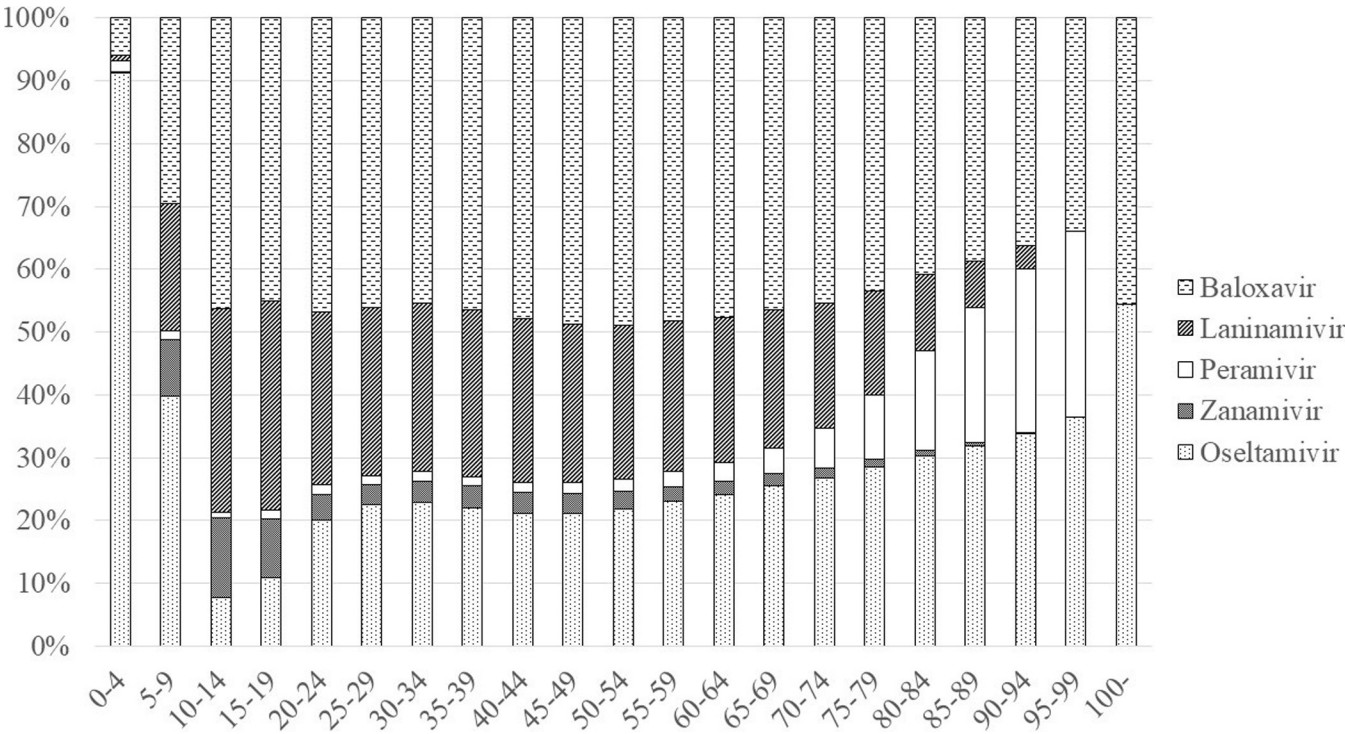

**Fig 5. Age and anti-influenza drugs in 2018.**

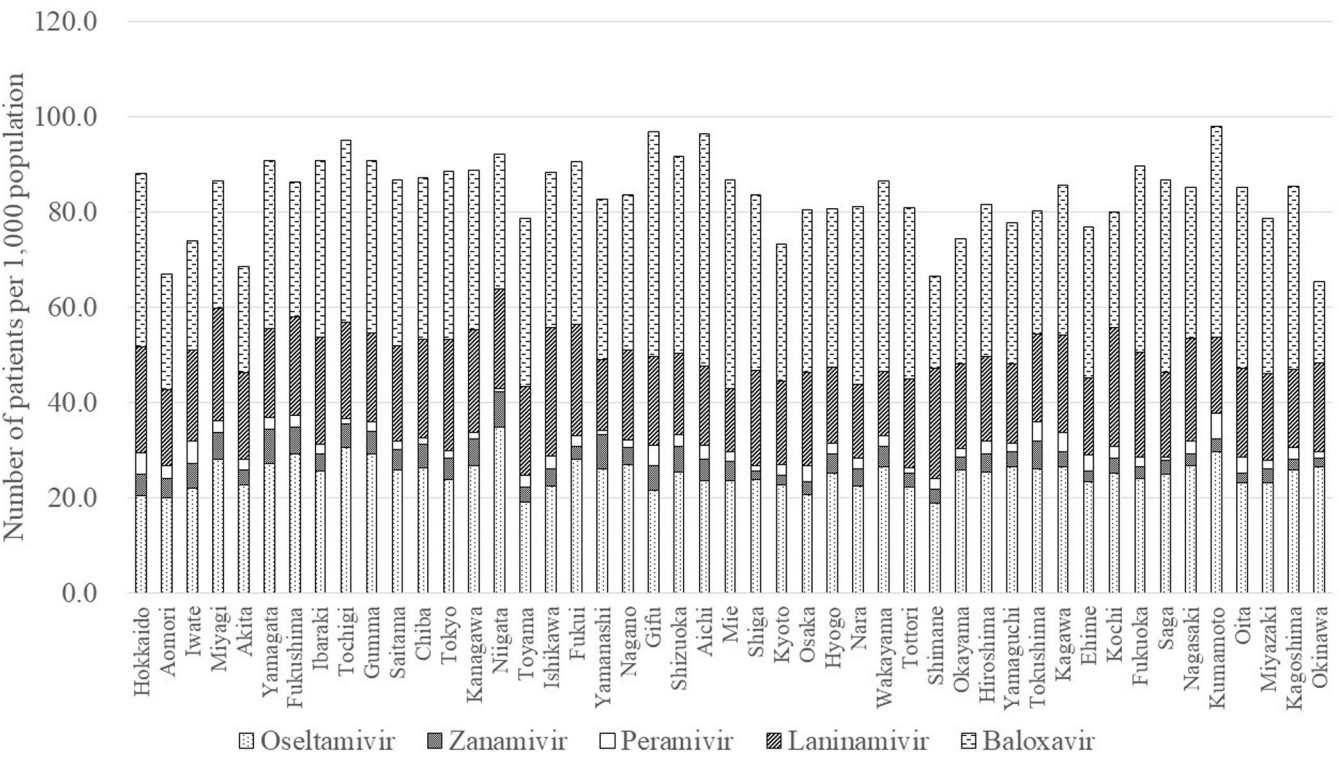

**Fig 6. Geographical differences by prefecture in patients prescribed anti-influenza drugs per 1,000 population in 2018.**

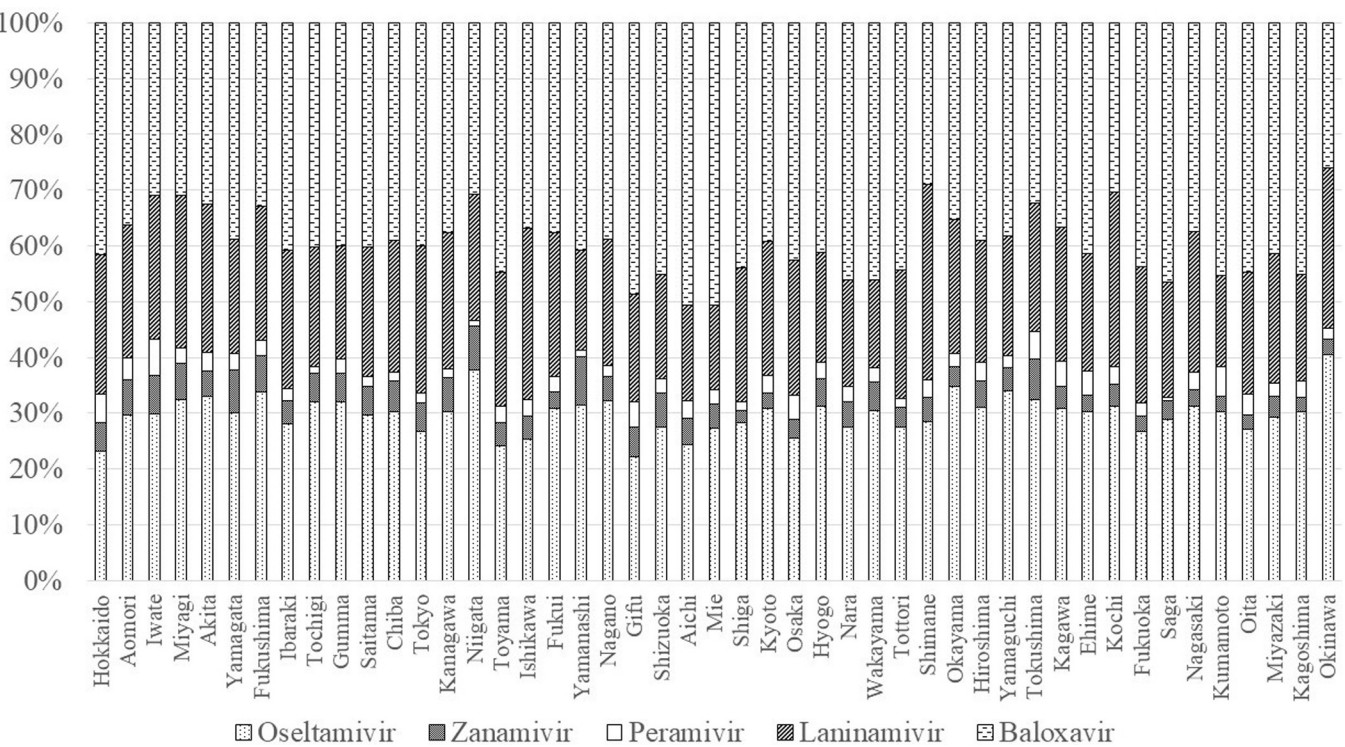

**Fig 7. Geographical differences by prefecture in prescriptions in 2018.**

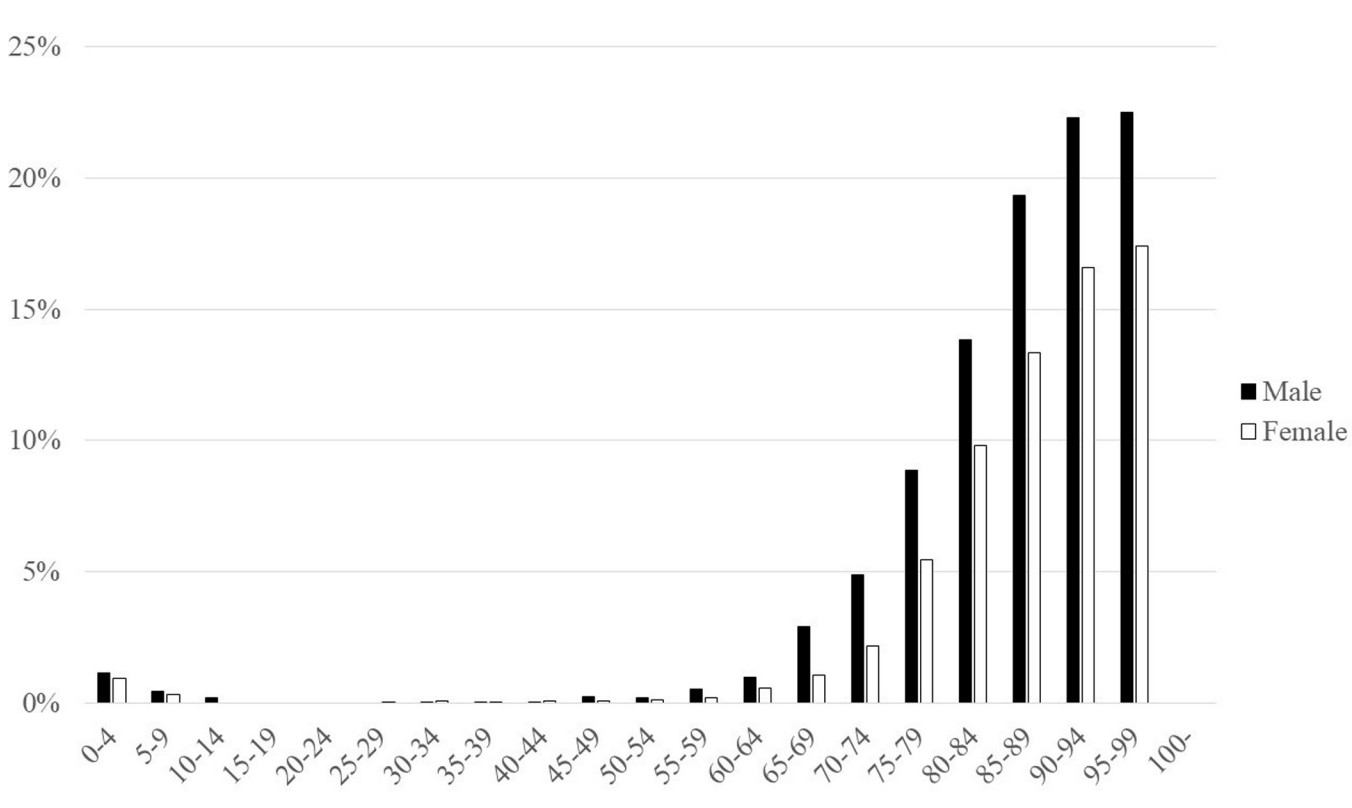

**Fig 8. Proportion of inpatients by age and sex in 2018.**

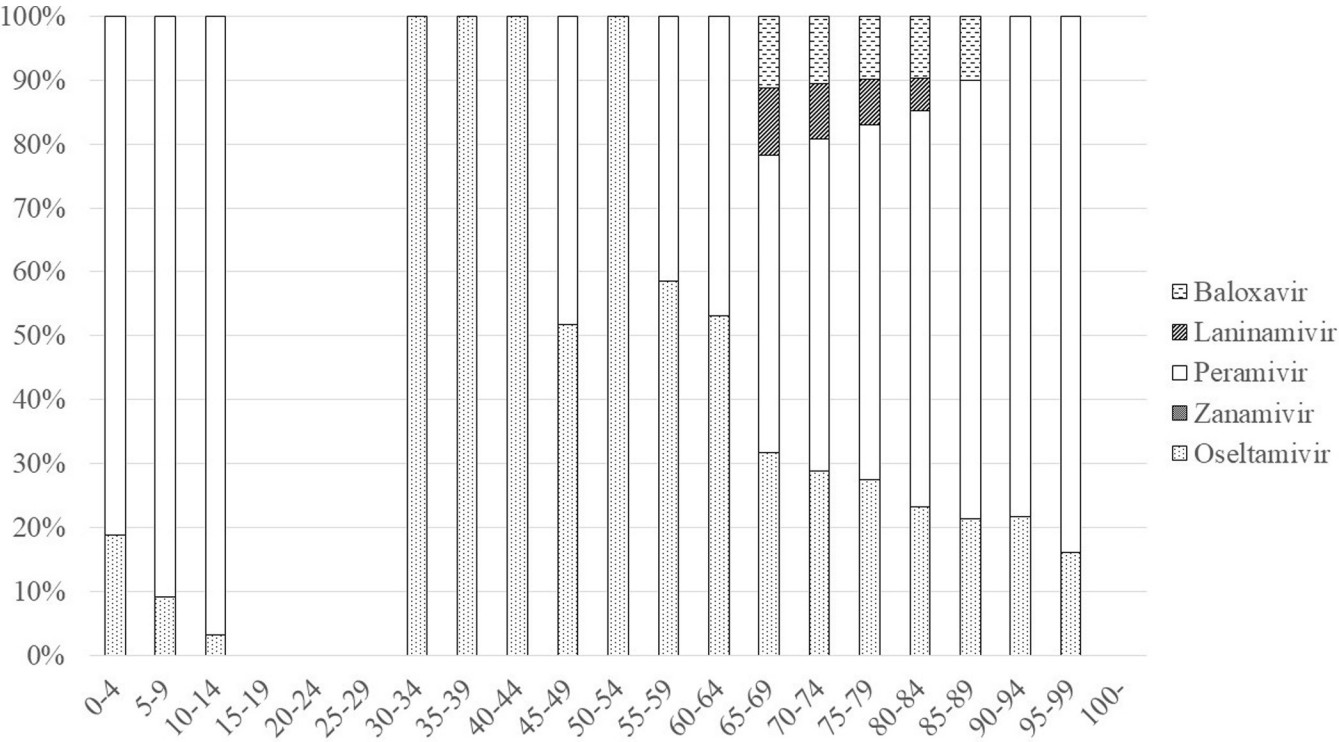

**Fig 9. Inpatient prescriptions for males in 2018.**

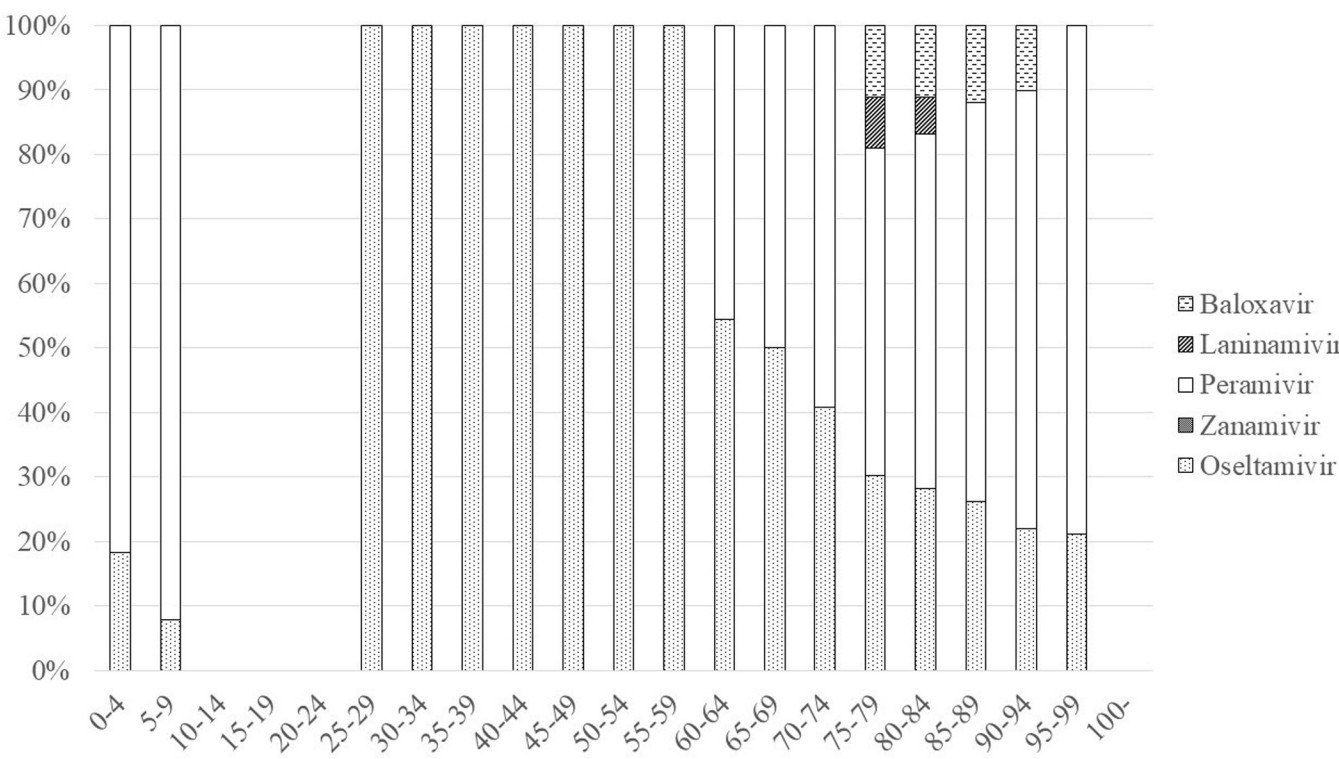

**Fig 10. Inpatient prescriptions for females in 2018.**

frequently prescribed anti-influenza drug for inpatients in Gifu (81.6%) and the least frequently prescribed in Okinawa (47.9%) (Fig 11).

## Discussion

This nationwide retrospective study investigated annual trends in anti-influenza drug use in Japan over the six-year period of 2014–2020. The estimated annual number of patients prescribed these drugs ranged from 6.7 to 13.4 million, while the costs ranged from JPY 30.1 to 47.1 billion. Patients aged < 20 years accounted for 46.1% of the anti-influenza prescriptions. The most frequently prescribed drug until 2017 was laninamivir, which was overtaken by the newly approved baloxavir in 2018, before prescriptions for baloxavir decreased dramatically the following year.

In Japan, the National Epidemiological Surveillance for Infectious Diseases (NESID) program is implemented nationwide [10,22,23]. The number of influenza patients estimated by NESID from the 2014–15 flu season to the 2019–20 flu season was 9.4, 9.8, 10.3, 14.4, 11.7, and 7.3 million, respectively, dropping to just 14,000 in the 2020–21 flu season. As defined by NESID, the flu season in Japan runs from the 36th week of the year (end of August or beginning of September) to the 35th week of the following year. Starting with the 2018–19 flu season, the estimation method was changed because the previous method resulted in overestimation (about 1.54 times) and the data prior to the 2017–18 season were corrected retrospectively. Our estimation of the annual number of patients prescribed anti-influenza drugs is similar to the number of influenza patients reported by NESID. This indicates that most influenza patients in Japan received an anti-influenza prescription. The age and sex distribution is also similar to that reported by NESID: patients aged 5–9 years received the largest proportion of prescriptions (17.1%) and patients aged < 20 and ≥ 60 years received 47.0% and 14.5%, respectively, and the ratio of male to female patients was 1:1. Both the NESID data and our data are mutually validated. In addition to our study, several other studies have investigated influenza by examining administrative claims databases, such as the Diagnosis Procedure Combination database [24,25], the Japan Medical Data Center database [26–28], and the NDB [29]. These databases are useful because they complement public health surveillance data, which sometimes lack clinical information such as prescriptions. In particular, the NDB is a nearly complete enumeration dataset. Secondary analyses of claims databases are less susceptible to underestimations due to underreporting and require less effort and cost compared with hospital- and physician-based surveillance.

It has been said that the management of influenza in Japan is unique, but analysis of large datasets has been lacking. Among studies conducted in Japan to date, an online survey of 200 households in Japan conducted in 2019 showed that 76% of participants with influenza-like illness visited a healthcare facility, 89% underwent a rapid influenza diagnostic test, and 95% who tested positive were treated with antivirals [5]. A study at a university hospital in France investigated 755 confirmed cases of influenza between 2018 and 2019, a quarter of which required hospitalization [30]. Oseltamivir was prescribed in 59.9% of cases, a rate the authors considered low and in need of improvement. A single-center study in Japan reported that prescription of anti-influenza drugs was associated with a positive influenza rapid antigen test, physician experience, and high regional influenza activity, whereas patient age and comorbidities were not associated [9]. The authors suggested the need for educational intervention because the evidence-based prescription of anti-influenza drugs was often neglected in clinical practice in Japan. Our nationwide data support the findings and concerns raised by these previous Japanese studies.

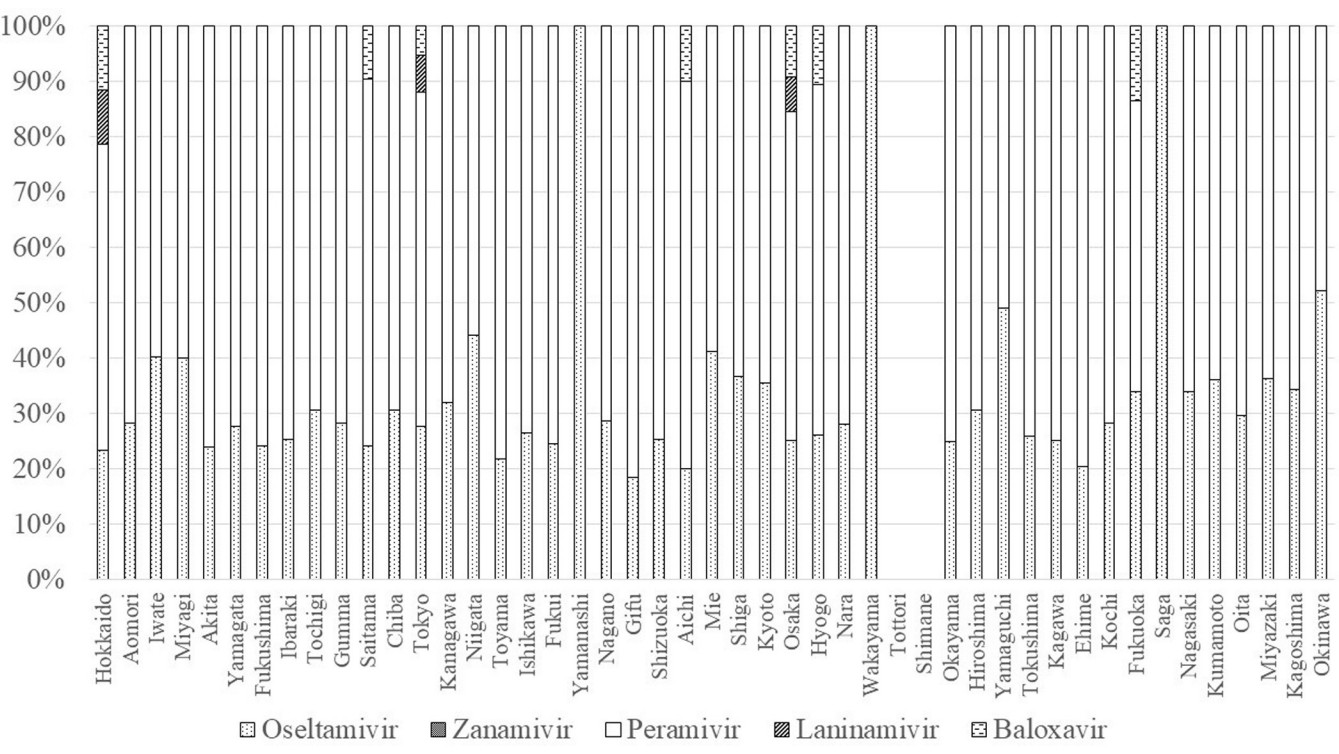

**Fig 11. Regional differences in inpatient prescriptions in 2018.**

Evidence for oseltamivir and zanamivir related to hard endpoints such as hospitalization and death had been questioned because systematic reviews had shown the efficacy of only soft endpoints such as time to alleviation [31–33]. However, in 2015, a systematic review of oseltamivir for documented influenza infection showed its efficacy in reducing lower respiratory tract infections and hospitalizations [34]. Recent US guidelines [35] are more proactive than before [36] in terms of anti-influenza drugs, not only for elderly and high-risk patients but also for otherwise healthy patients. In Japan, physicians seem to have routinely and proactively prescribed anti-influenza drugs regardless of the accumulation of evidence and updates to guidelines. Oseltamivir and zanamivir are the main anti-influenza drugs in the US and Europe [11,35]. Our analysis showed that laninamivir has been frequently prescribed in Japan, but evidence of its effectiveness is limited. Phase II trials in 12 countries failed to show its efficacy compared with placebo [37]. A randomized controlled trial in Japan showed non-inferiority to oseltamivir in terms of time to alleviation, and laninamivir has been approved only in Japan [38,39]. The emergence of variants with less susceptibility to baloxavir was reported in the first randomized controlled trial [40]. The Japan Pediatric Society previously did not recommend baloxavir for patients aged < 12 years because of the lack of evidence for children as well as the emergence of variants in 2018. It was only in 2018 that the Japanese Association for Infectious Diseases announced the characteristics of baloxavir [41]. Despite these concerns, baloxavir was the most prescribed anti-influenza drug in the 2018–19 flu season, just after it was released. In 2019, the Japanese Association for Infectious Diseases announced that they could not make a recommendation because of a lack of evidence. In the 2019–20 flu season, prescriptions of baloxavir dropped sharply. Although we could not determine why laninamivir and baloxavir are prescribed so frequently, Japanese physicians and patients might prefer the simplicity of single administration. It also seems that physicians prefer new drugs to conventional

drugs and that prescriptions were not driven by evidence or guidelines but rather by the promotion of new drugs by pharmaceutical companies [42,43]. Previous studies of cost-effectiveness did not comprehensively analyze all anti-influenza drugs in Japan and the results were inconclusive [44,45]. The generic form of oseltamivir became available in 2018, but we could not find any evidence that its availability influenced on physicians' prescription practice. Moreover, concerns about viral mutations related to resistance to neuraminidase inhibitors might have influenced their practice, although the relation with clinical ineffectiveness is uncertain [8]. Therefore, future studies should investigate how Japanese physicians decide which drugs to prescribe for influenza.

Our data showed a higher incidence of influenza in children. A quarter of the general population of children aged 5–9 years are prescribed anti-influenza drugs each year. There is a second peak in people in their 30s and 40s, with a higher incidence in women. This second peak may reflect the fact that some of these patients are childcare workers or the parents of children with influenza. In Japan, mothers spend much more time engaged in household and childcare work than fathers, according to a national survey on time use and leisure activities, and most nursery school and kindergarten teachers in Japan are women. A previous study that analyzed data from city-wide primary school seasonal influenza surveillance and a household survey showed that estimated contact intensity was higher in mother–child pairs than in father–child pairs [46]. It also showed that older patients, especially older men, were more likely to be hospitalized for influenza or to be infected with influenza during hospitalization. The in-hospital prescription rate for women in their 70s, 80s, and 90s was 3.6%, 11.4%, and 16.8%, respectively, and while it was higher for their male counterparts at 6.7%, 16.0%, and 22.3%. Boys were more likely to be prescribed anti-influenza drugs and their in-hospital prescription rate was higher than that for girls. It has been reported that males are more vulnerable to influenza in terms of risk of hospitalization and mortality compared with females [47]. In Hong Kong, male patients, especially those aged < 18 years, tended to have a higher excess hospitalization rate compared with female patients [48]. A retrospective single-center study in Canada that investigated the risk of hospitalization due to influenza in childhood showed that male sex was associated with a higher risk of hospitalization (adjusted odds ratio: 1.9; 95% confidence interval: 1.0–3.7) [49]. Our data are compatible with those of these previous studies, suggesting the need to take the patient's sex into consideration in the management of influenza.

Differences in the incidence of influenza by prefecture have been reported previously by NESID [22]. The present study is the first to show a geographical disparity in the choice of anti-influenza drugs prescribed. Previous studies of data from NDB Open Data have shown regional disparities in clinical practice in terms of surgery for nephrectomy and nephroureterectomy [50] as well as prescriptions for chronic kidney disease [51]. In addition to variance in the aging population and disease prevalence, the maldistribution of specialists might explain these disparities. However, influenza is a very common disease and most patients were treated by a pediatrician, internist, or general practitioner rather than an infectious disease or respiratory disease specialist. Establishment and dissemination of clinical guidelines is one of the most important factors in reducing disparities and standardizing clinical practice [52,53]. The Japan Pediatric Society has published clinical guidelines and recommendations about the choice of anti-influenza drugs, but there are no comprehensive clinical guidelines for seasonal influenza in adult patients, such as those published by the Infectious Diseases Society of America [35]. In addition, the Japanese Association for Infectious Diseases does not issue grade-based recommendations based on clinical evidence or cost-effectiveness [54]. It is therefore necessary to investigate how aware physicians are about the current guidelines and evidence for the effectiveness of anti-influenza drugs.

After the emergence of COVID-19, prescriptions for anti-influenza drugs decreased in 2019 and then dropped dramatically to just 14,000 in 2020. Although the significance of influenza was decreased, influenza is expected to re-emerge after restrictions to limit the spread of COVID-19 are lifted. Diagnosing COVID-19 depends largely on rapid antigen or polymerase chain reaction tests of nasopharyngeal swabs or saliva samples. Oral medications for COVID-19 such as molnupiravir and nirmatrelvir-ritonavir have recently become available. The impact of COVID-19 has decreased because of widespread vaccination, the emergence of less severe strains such as the Omicron variant, and the development of pharmacotherapies. Based on the similarity between influenza and COVID-19, we need to learn from both the strengths and shortcomings of influenza management in Japan in terms of clinical practice, logistics, and health policy.

This study has several limitations. First, NDB Open Data does not contain individual data but rather is aggregate data. The number of patients prescribed anti-influenza drugs can only be estimated, and the actual doses for pediatric patients and patients with renal impairment and the numbers of multiple prescriptions for individuals are not known. We could not perform a sensitivity analysis because of the lack of data available on the proportion of patients prescribed second-line anti-influenza drugs or the proportion of the patients with creatinine clearance 10–30 or < 10mL/min in Japan. However, mean body weight in children was derived from a national representative survey. Although we could not consider the dose adjustment of baloxavir for patients with a body weight $\geq$ 80 kg, proportion of the patients with a body weight $\geq$ 80 kg is not large because the mean ± standard deviation of body weight for Japanese men and women aged $\geq$20 years was reported to be 67.4 ± 12.0 kg and 53.6 ± 9.2 kg, respectively, by the National Health and Nutrition Survey in Japan in 2019 [55]. Based on a normal distribution, men prescribed baloxavir 80mg accounted for 14.7% at most. Second, patient characteristics other than age, sex, prefecture, inpatient/outpatient status, and outcome data are not available. Even if a patient's condition worsens and they are hospitalized the day after being prescribed anti-influenza drugs at an outpatient clinic, NDB Open Data only records that they were prescribed anti-influenza drugs as an outpatient. Third, as mentioned in the Methods section, the dataset contained missing values. However, these missing values accounted for only 0.2% of prescriptions, and the reason for the masked data is because of the small number of prescriptions in specific age categories and prefectures. Therefore, the influence of missing values is considered to be small. Fourth, there is a delay in data availability. NDB Open Data published the data for FY2020 in September 2022. Although these data are useful for public health surveillance, the delay in their publication makes it impossible to utilize them for timely management of seasonal influenza. Fifth, under the national health insurance scheme in Japan, reimbursements are made for anti-influenza drugs only when prescribed for treatment, not for prevention, and thus NDB contains only data on prescriptions for treatment. However, it cannot be ruled out that some prescriptions for prevention were included in the dataset and this may have resulted in overestimation of the number of patients. Sixth, the start and end points for each year in NDB Open Data and NESID differ. However, the influence of this difference on the annual estimate is considered to be small because case of influenza are uncommon from April to August in Japan, as shown by NESID [23].

Despite these limitations, our study clarifies the clinical epidemiology of prescription practices of anti-influenza drugs for seasonal influenza in Japan, where both physicians and patients are proactive toward management with anti-influenza drugs and rapid antigen tests. Our findings from nationwide real-world data can be utilized for public health management, health policy, and clinical practice not only in relation to influenza but also to COVID-19 and other emerging infectious diseases. Access to physicians, rapid antigen tests, and anti-influenza drugs are strengths of the Japanese health care system, but the choice of which anti-influenza

drug to prescribe remains unclear based on the clinical evidence currently available. Further research is needed to elucidate how physicians choose which anti-influenza drugs to prescribe.

## Author Contributions

**Conceptualization:** Akahito Sako.

**Data curation:** Akahito Sako.

**Formal analysis:** Akahito Sako, Yoshiaki Gu, Norio Ohmagari.

**Funding acquisition:** Akahito Sako.

**Investigation:** Akahito Sako.

**Methodology:** Akahito Sako.

**Project administration:** Hidekatsu Yanai.

**Supervision:** Yoshiaki Gu, Hidekatsu Yanai, Norio Ohmagari.

**Writing – original draft:** Akahito Sako.

**Writing – review & editing:** Akahito Sako, Yoshiaki Gu, Yoshinori Masui, Kensuke Yoshimura, Hidekatsu Yanai, Norio Ohmagari.

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
