## [Decision Letter · Decision Letter 0]

14 Mar 2023

PONE-D-23-02107Prescription of anti-influenza drugs in Japan, 2014-2020: a retrospective study using open data from the national claims databasePLOS ONE

Dear Dr. Sako,

Thank you for submitting your manuscript to PLOS ONE. After careful consideration, we feel that it has merit but does not fully meet PLOS ONE’s publication criteria as it currently stands. Therefore, we invite you to submit a revised version of the manuscript that addresses the points raised during the review process.

We look forward to receiving your revised manuscript.

Kind regards,

Jingjing Qian

Academic Editor

PLOS ONE

Journal Requirements:

Additional Editor Comments:

The cost analysis across years should be adjusted for inflation. For example, converting annual cost from other years to cost in 2020 using Japan's consumer price index in medical & pharmaceutical products. I am not sure if the authors incorporated this adjustment in their analysis. If not, please adjust your cost results.

Reviewers' comments:

Reviewer's Responses to Questions

**Comments to the Author**

1. Is the manuscript technically sound, and do the data support the conclusions?

Reviewer #1: Partly

Reviewer #2: Yes

2. Has the statistical analysis been performed appropriately and rigorously? 

Reviewer #1: No

Reviewer #2: Yes

3. Have the authors made all data underlying the findings in their manuscript fully available?

Reviewer #1: Yes

Reviewer #2: Yes

4. Is the manuscript presented in an intelligible fashion and written in standard English?

Reviewer #1: Yes

Reviewer #2: Yes

5. Review Comments to the Author

Reviewer #1: Review opinions are submitted as an attached document.

Review opinions are submitted as an attached document.

Review opinions are submitted as an attached document.

Review opinions are submitted as an attached document.

Reviewer #2: Dear authors,

Thank you for submitting your manuscript on the changes in influenza treatment prescriptions in Japan. While it is a descriptive study, it provides valuable insights into the use of influenza drugs and the availability of health insurance claim data in Japan.

Although the analysis is mainly descriptive, it may be worth considering the significance of the changes in drugs like oseltamivir. It would be useful to provide a discussion on the reasons behind these changes, such as new drugs and prices. Additionally, a cost-effectiveness evaluation of these drugs would provide a more comprehensive understanding of their value.

Overall, I agree that the use of health insurance claim data is an important and challenging aspect of the study. The regional significance of the results makes it a valuable contribution to the field. Therefore, I believe that it is possible to publish the manuscript in its present form.

Thank you for your contribution to the field of medicine, and I look forward to seeing your manuscript in print.

6. PLOS authors have the option to publish the peer review history of their article (what does this mean?). If published, this will include your full peer review and any attached files.

Reviewer #1: No

Reviewer #2: No

---

## [Author Response · Author response to Decision Letter 0]

15 Aug 2023

Responses to Reviewer/Editor Comments

Additional Editor Comments

The cost analysis across years should be adjusted for inflation. For example, converting annual cost from other years to cost in 2020 using Japan’s consumer price index in medical & pharmaceutical products. I am not sure if the authors incorporated this adjustment in their analysis. If not, please adjust your cost results.

Response: Thank you for raising this point. In Japan, consumer prices have been stable for decades. In fact, raising consumer prices and wages has recently become an issue of political, economic, and financial importance here. We feel that we do not have sufficient expertise to appropriately perform this adjustment for inflation. Therefore, we did not adjust for inflation, but instead show the crude drug costs. However, in addition to mentioning the price and the change in the price of anti-influenza drugs on page 9, we have added data from the consumer price index and mentioned that the government sets the price of drugs in Japan on page 10.

Reviewer 1

Influenza is an important infectious disease, and this study is considered to be a useful one. However, in the current manuscript, the methods presented by the authors did not seem to support the story of their study. Particularly, I feel that the most important point of this study, the method of estimating the number of patients based on the amount of anti-influenza drug prescriptions, is based on the authors' own speculation, and the authors have not sufficiently cited previous studies or conducted sensitivity analysis to verify the validity of their own method. On the other hand, the authors claim that the figures in this study are consistent with other studies in Japan. I also believe that there is a possibility of publication by giving full consideration to the significance, methods, and discussion of the study.

(BACKGROUND SECTION) 

1. There are few citations in the literature regarding the number of influenza cases in Japan and other countries. Does this mean that there are few prior studies on influenza incidence in Japan? Appropriate prior studies need to be cited. 

2. The clinical and international significance of this study is not clear. My question is: "What is the significance of counting patients?" Please specify what the academic significance of estimating the number of influenza patients and medical costs is. In the background section, you should describe how little is known from previous studies and what is newly proven in this study. 

Response: Thank you very much for your review. In response to the first two comments, the aim of our study was not to count the number of influenza patients but rather to elucidate prescription practices in Japan. We did not set out to compare the number of influenza cases in Japan with those in other countries. Therefore, we only refer to the number of influenza cases in Japan reported by the National Epidemiological Surveillance for Infectious Diseases (NESID). As far as we know, there is a lack of nationwide data on the prescription practices of anti-influenza drugs in Japan as well as in other countries. Aside from the data reported by NESID, no nationwide studies have investigated the incidence of influenza here in Japan. In response to your comment, we have revised the manuscript to make the focus of our study clearer.

3. (line 15 of p5) Related to #2, the justification for "there is a paucity of nationwide epidemiological and economic data" is not stated. Does this mean that The National Institute of Infectious Diseases reports data are not nation-wide? The reason for paucity is an important part of the significance of this study, so please describe it in detail. 

Response: The National Institute of Infectious Diseases reports nationwide data, which include annual estimates of influenza cases, deaths by influenza, seasonal trends, and distribution by region and age but not prescription data. Therefore, we investigated the clinical epidemiology of prescription practices for anti-influenza drugs. We have revised the Introduction section accordingly.

(METHODS SECTION) 

4. (line 11 of p6) Is "FY" an abbreviation that appears without explanation? Also, please describe the specific time period of FY. In a flu study, it is considered of most importance what month the statistics are from, and without that information, it is difficult to conduct proper peer review. (This relates to statistical adequacy.) 

Response:

We added text explaining that the fiscal year (FY) in Japan runs from April 1 to March 31. We have also added the NESID definition of “flu season” to the Discussion section, as well as the difference in the start and endpoints of the annual period between NDB Open Data and NESID to the limitations paragraph.

5. (line 15 of p6) I could not find a relation between this sentence and this study. Most of the papers cited have nothing to do with infectious diseases. 

Response:

In a database study, it is common to cite other studies that used the same database because it can help readers to better understand the database. It is also necessary to show the validity of studies that use the database, especially because NDB is not as well known as the Medicare and Medicaid database. Based on your comment, we have replaced some cited papers with papers related to infectious diseases.

6. (after line 2 of p7) The formula for converting the prescription volume of anti-influenza drugs to the number of patients is perhaps the most important point of this study. In particular, the number of pediatric patients cannot be ignored for influenza. Regarding children, whose weight gain is marked and individual differences are large, I could not determine whether your method of estimating the number of patients administered using national weight averages in 5-year increments was correct. It is essential to properly cite the demographic or pharmacoepidemiology literature to support this approach. I believe that a sensitivity analysis should be performed to examine the authors' assertions from multiple perspectives by varying the estimating equations. (This relates to statistical adequacy.) 

Response:

As described in the limitations paragraph, the actual weight and the actual prescribed dose are not available for pediatric patients in NDB Open Data. We have therefore newly cited data from the School Health Statistics and National Growth Survey on Preschool Children. If there is no national survey about children’s weight and different weight data are reported by multiple studies, we might need to perform a sensitivity analysis. However, we do not think that a sensitivity analysis is necessary in this case because we used nationally representative data. We could not find any other articles related to our approach. If we could have used NDB, we would have determined the actual dose for children and the precise number of patients prescribed. Therefore, we reported as much data from NDB Open Data as we could. We have added this point to the limitations paragraph. As a kind of sensitivity analysis, we have also added an estimation about baloxavir for patients weighing ≥ 80 kg.

7. (lines 14-20 of p7) Related to #6, the description of the validity of the authors' conversion formula prescription volume=>patient volume is lacking. The authors need to validate the conversion equation with some literature from pharmacoepidemiology or other sources. 

Response: 

Thank you for raising this point, but we did not develop an original conversion formula. Our estimation is simply based on the duration and dose written in the package instructions. We were not able to find any other pharmacoepidemiological studies or other resources to validate the conversion equation. However, we believe our estimation is adequate because it is considered that physicians would follow the instructions in the package insert. If physicians do not comply, the cost of the prescribed drug cannot be reimbursed. To clarify this point, we have added an explanation of the prescription and reimbursement process in Japan.

(RESULTS SECTION) 

8. The ranking of anti-influenza drug prescriptions is considered to be correct content. However, I read that it is what has been initially published in NDB Open Data and not the authors' original estimation, is this understanding correct? (Sorry if I am wrong.) 

Response: NDB Open Data contains just raw data about prescriptions, including those for anti-influenza drugs. Data on anti-influenza drugs include the brand and generic names, whether it was a capsule or dried syrup, whether the patient was an outpatient or inpatient, how the drug was administered (oral/inhalation/injection), and so on. From the NDB Open Data website, it is not easy to find a summary of the data on anti-influenza drug prescriptions such as a ranking of anti-influenza drugs. Open data in general are made publicly available so that they can be utilized by researchers. Accordingly, NDB Open Data does not report on the prescription practices of anti-influenza drugs in Japan as detailed in our manuscript, which is why we consider it worthwhile summarizing, analyzing, and publishing the findings for these publicly open clinical data. To clarify this matter, we have added an explanation about NDB Open Data on Page 7.

(DISCUSSION SECTION)

9. (p13) It is essential to prove the validity of the results of this study by examining in concrete detail the similarities with the results of the national survey conducted by NESID. (One of the most important points of this research.) 

Response: Thank you for your suggestion. We have added some of the concrete demographic data from NESID on page 14 in the Discussion section in order to compare and validate our study accordingly.

10. (lines 11-21 on p. 14) I understand that this description is relevant to the authors claim. However, how about stating the most important point above (proving the validity of the results of this study) in more detail? 

Response: We have revised the Discussion section and shortened the sentences accordingly.

11. (lines 1-8 on p15) How about a brief description? I feel that the clinical claims are strong compared to a weaker explanation of the validity of the results of this study. 

Response: We have revised the Discussion section and shortened the text accordingly.

Reviewer 2

Dear authors, 

Thank you for submitting your manuscript on the changes in influenza treatment prescriptions in Japan. While it is a descriptive study, it provides valuable insights into the use of influenza drugs and the availability of health insurance claim data in Japan.

Although the analysis is mainly descriptive, it may be worth considering the significance of the changes in drugs like oseltamivir. It would be useful to provide a discussion on the reasons behind these changes, such as new drugs and prices. Additionally, a cost-effectiveness evaluation of these drugs would provide a more comprehensive understanding of their value.

Overall, I agree that the use of health insurance claim data is an important and challenging aspect of the study. The regional significance of the results makes it a valuable contribution to the field. Therefore, I believe that it is possible to publish the manuscript in its present form.

Thank you for your contribution to the field of medicine, and I look forward to seeing your manuscript in print.

Response:

Thank you very much for your review. As suggested, we have added text on page 16 discussing the cost and cost-effectiveness of the drugs as well as concerns about resistance to oseltamivir, which might have influenced physicians’ prescription practices. On page 18, we have added a sentence stating that the Japanese guidelines do not mention cost-effectiveness.

---

## [Decision Letter · Decision Letter 1]

4 Sep 2023

Prescription of anti-influenza drugs in Japan, 2014-2020: a retrospective study using open data from the national claims database

PONE-D-23-02107R1

Dear Dr. Sako,

We’re pleased to inform you that your manuscript has been judged scientifically suitable for publication and will be formally accepted for publication once it meets all outstanding technical requirements.

Kind regards,

Jingjing Qian

Academic Editor

PLOS ONE

Additional Editor Comments (optional):

Thanks for addressing comments from both reviewers from the original review. And thanks for providing the rationale of no cost adjustment for your analysis and results.

Reviewers' comments:

Reviewer's Responses to Questions

**Comments to the Author**

1. If the authors have adequately addressed your comments raised in a previous round of review and you feel that this manuscript is now acceptable for publication, you may indicate that here to bypass the “Comments to the Author” section, enter your conflict of interest statement in the “Confidential to Editor” section, and submit your "Accept" recommendation.

Reviewer #1: All comments have been addressed

Reviewer #3: (No Response)

2. Is the manuscript technically sound, and do the data support the conclusions?

Reviewer #1: (No Response)

Reviewer #3: Yes

3. Has the statistical analysis been performed appropriately and rigorously? 

Reviewer #1: (No Response)

Reviewer #3: Yes

4. Have the authors made all data underlying the findings in their manuscript fully available?

Reviewer #1: (No Response)

Reviewer #3: Yes

5. Is the manuscript presented in an intelligible fashion and written in standard English?

Reviewer #1: (No Response)

Reviewer #3: Yes

6. Review Comments to the Author

Reviewer #1: (No Response)

Reviewer #3: I appreciate great efforts of the authors in addressing reviewers’ comments. Given the limitation of the data, I believe the findings of the study may significantly contribute to current literature and invite further investigation of anti-influenza medication prescriptions in Japan. After incorporating reviewers’ comments, I think the quality of content in the manuscript improved. However, there are some remaining points that the authors can take into consideration.

1. The Methods section seems to be lengthy and contains too much text. In describing the anti-influenza drugs, the authors can create a table for all available anti-influenza medications, with marketed years, indications, dosing, dosage forms, and their pediatric use. In addition, the authors should describe the methods to identify these drugs in NDB Open Data (i.e., by search string or NDC/ATC code, etc) and the validity of exposure ascertainment. Since the authors mentioned the presence of missing data in NDB, this can also pose a threat to the estimates.

2. The authors organized the Methods section by describing the components of the NDB Open Data rather than focusing on the components of an observational study. Please refer to the STROBE guidelines to describe the components of the study, in terms of study design, data sources, exposure, outcome/measure, covariates, statistical analysis, and sensitivity analysis. This can make the manuscript more attractive to the readers.

3. From statistical perspective, any estimation should have uncertainty. However, I could not find any uncertainty of estimates (i.e., 95% confidence intervals, standard errors). Could the authors clarify more on the estimate of annual number of prescriptions/tests or cost in the analysis?

7. PLOS authors have the option to publish the peer review history of their article (what does this mean?). If published, this will include your full peer review and any attached files.

Reviewer #1: No

Reviewer #3: No

---

## [Editor Report · Acceptance letter]

11 Sep 2023

PONE-D-23-02107R1 

Prescription of anti-influenza drugs in Japan, 2014-2020: a retrospective study using open data from the national claims database 

Dear Dr. Sako:

I'm pleased to inform you that your manuscript has been deemed suitable for publication in PLOS ONE. Congratulations! Your manuscript is now with our production department. 

Kind regards, 

on behalf of

Dr. Jingjing Qian 

Academic Editor

PLOS ONE